# [1]Ferrocenophane Bridged by a 9-Silafluorenylidene Moiety

Shinnosuke Usuba [1], Koh Sugamata [2], Shogo Morisako [3,†] and Takahiro Sasamori [1,3,4,*]

1 Graduate School of Science and Technology, University of Tsukuba, 1-1-1 Tennoudai,
Tsukuba 305-8571, Ibaraki, Japan; usuba@dmb.chem.tsukuba.ac.jp
2 Department of Chemistry, College of Science, Rikkyo University, 3-34-1 Nishi-Ikebukuro, Toshima-ku,
Tokyo 171-8501, Japan; sugamata@rikkyo.ac.jp
3 Division of Chemistry, Institute of Pure and Applied Sciences, University of Tsukuba, 1-1-1 Tennoudai,
Tsukuba 305-8571, Ibaraki, Japan; s-morisako@sagami.or.jp
4 Tsukuba Research Center for Energy Materials Sciences (TREMS), University of Tsukuba, 1-1-1 Tennoudai,
Tsukuba 305-8571, Ibaraki, Japan
* Correspondence: sasamori@chem.tsukuba.ac.jp; Tel.: +81-29-853-4412
† Current address: Sagami Chemical Research Institute, Hayakawa 2743-1, Ayase 252-1193, Kanagawa, Japan.

**Abstract:** Sila[1]ferrocenophane bearing a 9-silafluorenylidene moiety (**1**) as a bridging unit was synthesized and isolated as a stable crystalline compound. Sila[1]ferrocenophane **1**, which was newly obtained in this study, was characterized by spectroscopic analyses, a single-crystal X-ray diffraction (SC-XRD) analysis, and electrochemical measurements. Due to the characteristic 9-silafluorenyl moiety, **1** exhibited large electron affinity and a slightly higher oxidation potential relative to that of ferrocene. In addition, **1** was found to undergo ring-opening polymerization (ROP) triggered by thermolysis at a lower temperature relative to that of $Ph_2Sifc$ (**1′**, fc = 1,1′-ferrocenylidene). It also underwent ROP through reduction by $KC_8$ to give the corresponding polymeric compound. The DFT calculations suggested that one-electron reduction of **1** would promote ring-opening polymerization, as shown in the experimental results.

**Keywords:** sila[1]ferrocenophane; 9-silafluorenylidene; ring-opening polymerization; redox behavior; cyclic voltammetry

## 1. Introduction

There has been much interest in the chemistry of sila[1]ferrocenophanes, which are [1]ferrocenophane derivatives bearing a single silicon atom as a bridging unit between the two Cp rings of the ferrocenyl moiety [1,2], because they could be easily converted to Si/Fe containing metallopolymers bearing several types of functional moieties on silicon atoms via anionic, cationic, transition-metal-catalyzed, or thermal ring-opening polymerization (ROP) [1–6]. Sila[1]ferrocenophanes have attracted much attention in main group element chemistry since the first isolation of highly strained $Ph_2Sifc$ (**1′**, fc = 1,1′-ferrocenylidene) not only as suitable precursors of a metallopolymer, but also as highly strained silicon molecules [7]. Particularly, the electronic interactions between the ferrocenyl and silyl moieties should be of great interest, because the Fe atom and the silicon moiety would be forced to be close to each other along with severe strain to yield several orbital overlaps. Here, we are interested in the synthesis of sila[1]ferrocenophane **1** bearing a 9-silafluorenylidene unit from the viewpoint of the electronic interaction between the electron donating a 1,1′-ferrocenylidene unit and the electron-accepting moiety of a 9-silafluorenylidene unit [8–10]. It should especially be of great interest that a 9-silafluorenylidene moiety is highly electrophilic due to the aromatic contribution of the resulting anion species besides the electron-donating ability of a ferrocenyl moiety. Thus, the combination of a ferrocenyl moiety and a 9-silafluorenylidene could behave as a unique donor-acceptor system. In addition, the polymer which would be formed by the ROP of **1** should be of great interest because of the possible alternate arrangement of ferrocenylidene and 9-silafluorenylidene units, since the

polymeric compounds of 9-silafluorenylide species have attracted much attention [8,11,12]. Its strained skeleton and the possible stability of the corresponding anionic species should promote its anionic or ring-opening polymerizations.

## 2. Results and Discussion

According to the reported procedure [13], 9,9-dichloro-9-silafluorene (**2**) was synthesized by the treatment of $SiCl_4$ with 2,2′-dilithiobiphenyl, which was generated by the reaction of 2,2′-dibromobiphenyl with *n*-BuLi. The reaction of **2** with 1,1′-dilithioferrocene·(tmeda)$_2$ [14] (tmeda = tetramethylethylenediamine) resulted in the formation of sila[1]ferrocenophane **1** as an orange solid in 67% isolated yield (Scheme 1, Figures S1–S3). Sila[1]ferrocenophane **1** is a stable crystalline compound in the air, but it undergoes slow decomposition in a polar solvent such as tetrahydrofuran (THF) or $CH_2Cl_2$.

**Scheme 1.** Synthesis of sila[1]ferrocenophane **1**.

The molecular structure of **1** has been revealed by the single-crystal X-ray diffraction analysis (SC-XRD) (Figure 1) [15]. Since the structural parameters of 9-silafluorenylidene-bridged [1]ferrocenophane have been revealed for the first time, they should be compared with those of other sila[1]ferrocenophane derivatives, and the comparison of structural features between **1** and diphenylsilyl-bridging [1]ferrocenophane (**1′**) [7] are especially worth discussing. As one can see from the molecular geometry of **1**, its ferrocenylidene moiety and the 9-silafluorenyl skeleton are almost perpendicularly located, where the dihedral angle of $C_1SiC_6$ and 9-silafiluorenylidene planes is 84.9°. In addition, the packing structure of **1** showed the intermolecular parallel ordering of the 9-silafluorenylidene moieties with a 3.41 Å distance, which was probably due to the π stacking macro structure. In the consideration of the evaluative geometric parameters for the skeletal strain in a sila[1]ferrocenophane (α, β, θ, δ in Figure 1c) [1,2], the skeletal strain in **1** as a sila[1]ferrocenophane should be almost the same as that in the SiPh$_2$-bridged sila[1]ferrocenophane (**1′**, α = 19.1°, β = 40.1°, θ = 99.2°, δ = 167.3°) [7], and slightly smaller relative to that in the Si(CH$_3$)$_2$-bridged sila[1]ferrocenophane (**1″**, α = 20.8°, β = 37.6°, θ = 95.7°, δ = 164.7°) [10]. Notably, the Si–C bond lengths in the strained ferrocenylidene-9-silafluorenylidene moiety (Si–C1, 1.884(3) Å, Si–C6, 1.888(3) Å) are slightly longer than those in the 9-silafluorenyl moiety (Si–C11, 1.859(2) Å, Si–C22, 1.854(2) Å), which is probably due to the hyper-conjugation between the Si–C(ferrocenylidene) σ* and π(9-silafluorenyl moiety) orbitals, indicating that the bridging Si–C σ bonds would be weakened by the severe strain of [1]ferrocenophane and the hyper conjugation of a 9-silafluorenyl derivative. Indeed, the Si–C(ferrocenylidene) lengths in the Si(CH$_3$)$_2$-bridged [1]ferrocenophane (**1″**) are reported to be 1.851(9) and 1.865(9) Å [10], which are slightly shorter relative to those in **1**, probably due to the lack of hyper-conjugation including Si–C σ* orbitals. While the structural parameters of **1** seem to be almost similar to those of **1′** (Si–C(fc), 1.881(11) Å), the quality of the reported SC-XRD data seem to be insufficient to discuss the comparison in detail [7].

The theoretically optimized structure of **1** at the B3PW91-D3(BJ)/6-311G(3d) level exhibits the characteristic structural parameters similar to those experimentally observed in the SC-XRD analysis (Si–C(ferrocene) = 1.878 Å, Si–C(fluorenylidene) = 1.858 Å, Fe–Si, 2.669 Å), suggesting negligible perturbation in the structural parameters due to the packing force in crystals. The evaluative geometric parameters for the skeletal strain in the theoretically optimized structure of **1** are α = 19.3°, β = 39.1°, θ = 97.0°, and δ = 165.7°, also suggesting the negligible perturbation by the packing force and reliability in the levels of the density functional theory (DFT) calculations in these systems. Notably, the natural

bond orbital (NBO) calculations of **1** indicated the small contribution of the donor–acceptor type interaction between the Fe moiety (donor) and 9-silafluorenilydene moiety (acceptor) as ca. 2 kcal/mol based on the second-order perturbation theory.

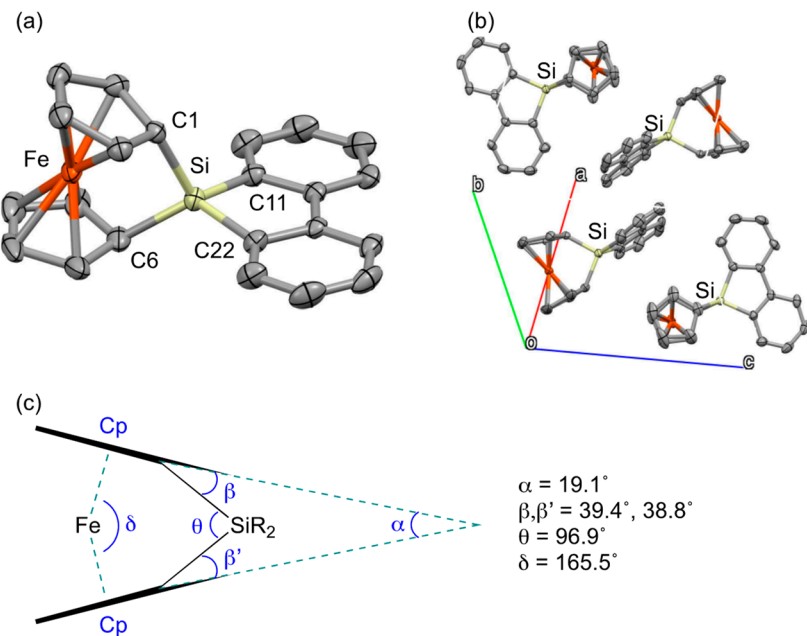

**Figure 1.** (**a**) Molecular structure and (**b**) packing structure in the unit cell of **1** (ORTEP drawing, 50% probability) and (**c**) geometric parameters of **1**. Selected atomic distances (Å): Si–C1,1.884(3), Si–C6, 1.888(3), Si–C11, 1.859(2), Si–C22, 1.854(2), and Fe–Si, 2.6817(8).

The UV/vis spectra of **1** in THF showed characteristic absorptions at $\lambda_{max}$ = 277 nm ($\varepsilon$ = 1.2 × 10$^4$), 476 nm ($\varepsilon$ = 1.9 × 10$^2$). The time-dependent DFT (TD-DFT) calculations at the TD-B3PW91-D3(BJ)/6-311G(3d) level suggested that the assignment of the absorption at a shorter wavelength ($\lambda_{max}$ = 277 nm) should be assignable to the $\pi$-$\pi$* electron transitions in the 9-silafluorenyl moiety (calculated at $\lambda$ = 296 nm). While the HOMO and LUMO of **1** should be localized on the ferrocenyl and 9-silafluorene moieties, respectively, as shown in Figure 2a,b, the second absorption at the longer wavelength ($\lambda_{max}$ = 476 nm) could be due to the mixture of the HOMO-LUMO electron transitions and the d-d electron transitions in the ferrocenyl moiety (calculated at $\lambda$ = 473, 470, 456, 430 nm). Notably, the contribution of HOMO-LUMO electron transitions would be negligible in the observed absorptions, because the UV/vis spectra in THF and benzene ($\lambda_{max}$ = 479 nm, $\varepsilon$ = 3.7 × 10$^2$) were almost the same. While the geometries of SiR$_2$ moieties in 9-silafluorenylidene-bridged sila[1]ferrocenophane **1** and diphenylsilyl-bridged sila[1]ferrocenophane **1′** are totally different from each other around the aryl groups on the Si atom (Figure 2), the locations of the HOMO and LUMO are similar, i.e., the HOMO and LUMO of **1′** are also located on the ferrocenyl and Ph$_2$Si moieties, respectively. The LUMO level of **1** should be considerably lower than that of the SiPh$_2$-bridged sila[1]ferrocenophane **1′** (LUMO: −0.95 eV), while the HOMO levels of **1** (−5.75 eV) and **1′** (−5.71 eV) would be almost the same, reflecting the hyper-conjugation system of the 9-silafluorenylidene moiety due to $\pi$-$\sigma$*(Si) conjugation.

Furthermore, based on the theoretical calculations at the B3PW91-D3(BJ)/6-311G(3d) level, the strain energies in fc(SiR$_2$) (fc = 1,1′-ferrocenylidene, R$_2$ = biphenylidene (**1**), and Ph$_2$ (**1′**)) were estimated according to the isodesmic reaction schemes shown in Scheme 2, which is the ideal reaction of the strained sila[1]ferrocenophane with phenylsilane giving the corresponding 1′-(SiH$_3$)-1-(R$_2$PhSi)-ferrocene. As a result, the calculated strain energy of **1** (33.1 kcal/mol) should be slightly larger than that of the SiPh$_2$-bridged sila[1]ferrocenophane (**1′**, 27.8 kcal/mol). In the consideration of these theoretical results,

it can be thought that **1** could undergo ring-opening polymerization with a lower barrier relative to that of **1′**.

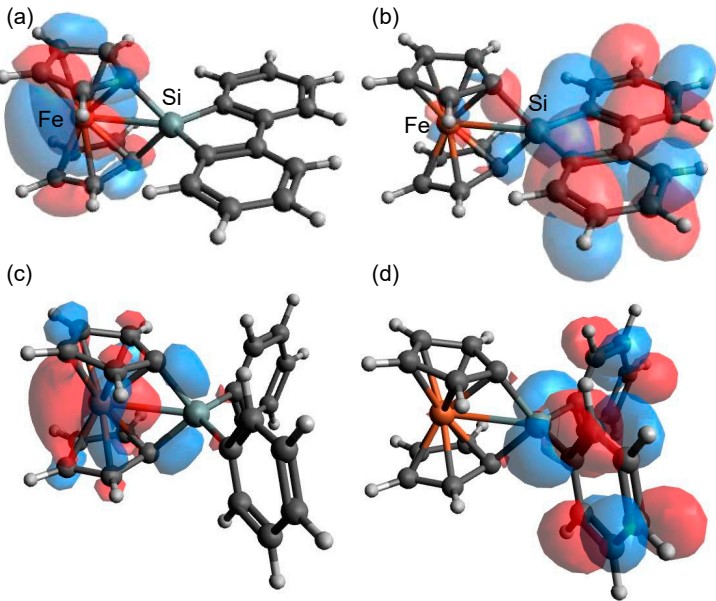

**Figure 2.** (**a**) HOMO (−5.75 eV) and (**b**) LUMO (−1.43 eV) of 9-silafluorenylidene-bridging sila[1]ferrocenophane **1**. (**c**) HOMO (−5.71 eV) and (**d**) LUMO (−0.95 eV) of diphenylsilyl bridging sila[1]ferrocenophane **1′**.

**Scheme 2.** Isodesmic reactions of sila[1]ferrocenophanes **1** and **1′**.

The obtained sila[1]ferrocenophane **1** is stable in the air. Heating sila[1]ferrocenophane **1** in the solid state under a reduced pressure at 190 °C resulted in orange solids (Scheme 3). The detailed identification of the obtained orange solids was very difficult due to its extremely poor solubility in common organic solvents such as THF, CHCl$_3$, and CH$_2$Cl$_2$. A very small amount of the orange solid, however, could be solved in C$_6$D$_6$ for the observation of the $^1$H NMR spectrum, which showed the broadened signals due to biphenyl moieties and ferrocenyl moieties. Thus, it can be concluded that **1** underwent thermal ROP to produce the corresponding polymer **3** as insoluble orange solids at a slightly lower temperature (190 °C) relative to that of **1′** (230 °C) [6], although the detailed analysis of the polymerization temperature has not been performed at this stage.

Because of its redox-active ferrocenyl substituents, the electrochemical property of **1** should be of great interest, and this was investigated by cyclic voltammetry (CV) and differential potential voltammetry (DPV) at room temperature in CH$_2$Cl$_2$ (Figure 3). The cyclic voltammogram of **1** showed a pseudo-reversible one-step redox couple in the oxidation region at $E_{1/2}$ = 0.34 V (vs. FcH/FcH$^+$). The higher oxidation potential of **1** relative to ferrocene may be explained by the low-lying $\pi^*$ orbital of the 9-silafluorenylidene moiety (Figure 2) [11,12]. In the reduction region, both cyclic and differential pulse voltammograms measured in THF and CH$_2$Cl$_2$ did not show a clear peak but showed gradual current waves, indicating that a continuous electron transfer would occur to produce decomposed products. Although we have attempted several measurements at a lower temperature

and/or in several types of solvents, the results were almost the same. Thus, we thought **1** underwent irreversible reduction at the electrode to produce a decomposed product.

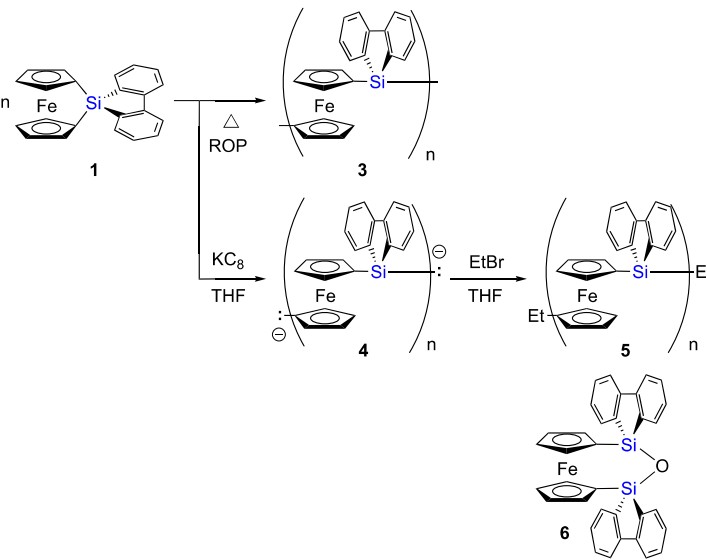

**Scheme 3.** Ring-opening polymerization of sila[1]ferrocenophane **1**.

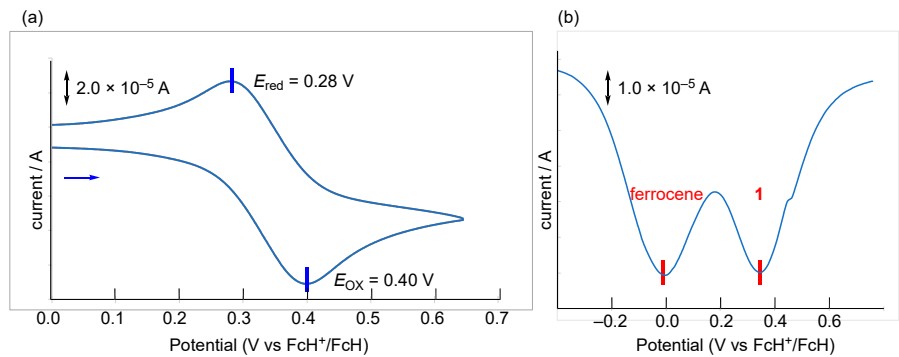

**Figure 3.** (**a**) Cyclic voltammogram and (**b**) differential pulse voltammogram * of **1** in $CH_2Cl_2$ at room temperature (2.0 mM, 0.1 M [$n$Bu$_4$N$^+$] [PF$_6{}^-$]); 0.05 V/s. * An equimolar amount of ferrocene was included in the solution as a standard.

Indeed, the chemical reduction of **1** with KC$_8$ in THF afforded the dark black suspension. Quenching the reaction mixture with an electrophile, such as EtBr, resulted in orange solids, which exhibit very low solubility in organic solvents as well as polymer **3**, which make its identification very difficult. In the consideration of the similarity in the $^1$H NMR spectrum of **5** to that of **3**, the obtained orange solids in the reduction of **1** followed by the addition of EtBr should be ethylated polymer **5**. Accordingly, the dark black intermediate observed in the reduction of **1** should be dianionic polymer **4**, although its identity has not been unambiguously established at present. Thus, it can be concluded that the reduction of **1** would promote its anionic ROP to give the corresponding dianionic polymer **5**, which would give the alkylated derivative by the addition of alkyl halide. The theoretical calculations suggested that the electron affinity of **1** should be estimated as positive (0.37 eV) in contrast to the negative electron affinity of **1′** (−0.08 eV), supporting the facile anionic ROP in the case of **1**. It should be noted that the theoretically optimized structures of anion radical **1**·⁻ showed considerable structural deformation relative to the corresponding neutral species **1** ($\alpha$ = 22.4°, $\beta$ = 34.5°, $\theta$ = 90.6°, $\delta$ = 162.4°), suggesting facile anionic ring-opening polymerization. However, the one-electron reduction of **1′** would cause only a small change in the structural deformation (**1′**·⁻, $\alpha$ = 18.9°, $\beta$ = 38.6°, $\theta$ = 87.0°, $\delta$ = 165.3°) in contrast to the case of **1** as described above.

In addition, during the anionic ROP process of **1**, [3]ferrocenophane **6** (Figure 4) with a disiloxane moiety was partially obtained, which was probably due to the contamination of air and moisture. In consideration of the structure of **6** bearing two silafluorenyl moieties per one ferrocenyl unit, it might be considered that the intermediary oligomeric compound of the 9-silafluorenylidene-ferrocenylidene moieties would undergo hydrolysis by the contaminated $OH^-$ ion under the reduction conditions to obtain **6**. That is, **6** could be generated through the hydrolysis of the $-(C_{10}H_8Fe)-(SiC_{12}H_8)-(C_{10}H_8Fe)-(SiC_{12}H_8)-(C_{10}H_8Fe)-$ moiety to obtain $-(C_{10}H_8Fe)-H$ $HO-(SiC_{12}H_8)-(C_{10}H_8Fe)-(SiC_{12}H_8)-OH$ $H-(C_{10}H_8Fe)-$, the middle part of which would undergo intramolecular dehydration to obtain **6**. While this hypothesis lacks strong evidence, the occasional formation of **6** might indicate the polymeric structure of **4** and **5** because **6** has two 9-silafluorenyl moieties attached to the ferrocenyl moiety. The molecular structure of **6** was definitively revealed on the basis of spectroscopic and SC-XRD analyses (Figure 4). This is the first example of the XRD analysis of **6** as a $-Si(C_{12}H_8)-O-Si(C_{12}H_8)-$ bridged [3]ferrocenophane derivative. As in the case of the previously reported $-SiR_2-O-SiR_2-$ bridged [3]ferrocenophanes (R = $CH_3$, Ph) [16,17], **6** was found to exhibit a negligible strain since the tilt angle between the two Cp planes of **6** is 1.18° (while those of the $-SiR_2-O-SiR_2-$ bridged [3]ferrocenophanes are 0.45° (R = Me) and 2.5° (R = Ph)). In addition, the Si–O–Si plane is located almost vertically towards one of the Cp ring (*ca.* 81°) but somehow tilted from another Cp plane (*ca.* 62°). That is, the two Cp rings are twisted around each other, and the Si–O–Si linkage is also twisted somehow. While packing structure of **1** in the single crystal shows π-π stacking intermolecular interactions between the 9-silafluorenylidene moieties, the packing structure of **6** in the single crystal is mostly explained by the CH–π intermolecular interaction between the 9-silafluorenylidene moieties and the π-π stacking interactions between the 9-silafluorenylidene moiety and the cyclopentadienyl ring of the ferrocenyl unit.

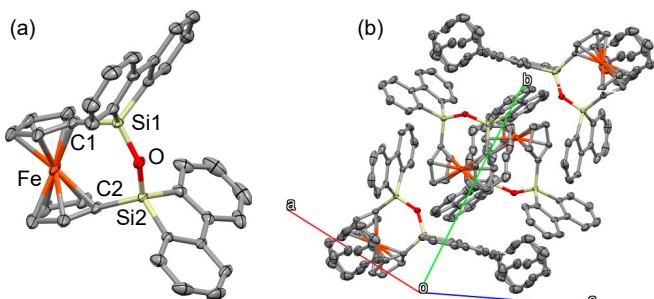

**Figure 4.** (**a**) Molecular structure and (**b**) packing structure in the unit cell of **6** (ORTEP drawing, 50% probability). Toluene molecules included in the single crystal were omitted for clarity. Selected atomic distances (Å) and angles (°): Si1–C1, 1.849(1); Si2–C2, 1.857(1); Si1–O, 1.644(1); Si2–O, 1.643(1); and Si1–O–Si2, 139.61(6).

## 3. Materials and Methods

### 3.1. General Information

All manipulations were carried out under an argon atmosphere using Schlenk-line techniques. All solvents were purified using standard methods. Trace amounts of water and oxygen remaining in the solvents were thoroughly removed by bulb-to-bulb distillation from potassium mirror prior to use. $^1H$, $^{13}C$, and $^{29}Si$ NMR spectra were measured on a Bruker AVANCE-400 spectrometer ($^1H$: 400 MHz; $^{13}C$: 101 MHz; $^{29}Si$: 79.5 MHz, Bruker Japan Co., Ltd., Kanagawa, Japan). Signals arising from residual $C_6D_5H$ (7.16 ppm) in $C_6D_6$ was used as the internal standard for the $^1H$ NMR spectra, that of $C_6D_6$ (128.0 ppm) was used for the $^{13}C$ NMR spectra, and external $SiMe_4$ (0.0 ppm) was used for the $^{29}Si$ NMR spectra, respectively. Multiplicity of signals in $^{13}C$ NMR spectra was determined using the DEPT technique. High-resolution mass spectra (HRMS) were obtained on a JEOL JMS-T100LP (DART) mass spectrometer (JEOL Ltd., Tokyo, Japan). All melting points were

determined on a Büchi Melting Point Apparatus M-565 (Nihon BUCHI K.K., Tokyo, Japan) and are uncorrected.

### 3.2. Synthesis of 9,9-Dichloro-9-silafluorene (**2**) [13]

2,2′-Dibromobiphenyl (3.39 g, 10.9 mmol) was dissolved in THF (33.0 mL) in a round flask and cooled at −78 °C. After a hexane solution of *n*-BuLi (8.4 mL, 2.69 M in hexane, 21.8 mmol) was added dropwise to the flask, the solution was stirred at −78 °C for 2 h. Then, the resulting yellow suspension was added to a THF solution of tetrachlorosilane (6.09 mL, 54.5 mmol) at −98 °C, and it was stirred for 20 h and slowly warmed to room temperature. All volatiles were removed under reduced pressure, and $Et_2O$ was added. After all insoluble inorganic salts were removed by filtration through a pad of celite, the solvent was removed from the filtrate under reduced pressure to obtain 9,9-dichloro-9-silafluorene (**2**) as colorless solids (2.65 g, 10.5 mmol, 96%). The spectral and analytical data were identical to those reported in the literature [13].

### 3.3. Synthesis of Sila[1]ferrocenophane **1**

Ferrocene (495 mg, 2.66 mmol) was dissolved in hexane (5.0 mL) in a J Young Schlenk bottle. After a hexane solution of *n*-BuLi (2.1 mL, 2.66 M in hexane, 5.59 mmol) and *N,N,N′,N′*-tetramethylethylenediamine (tmeda, 0.84 mL, 5.64 mmol) was added dropwise to the Schlenk bottle, the solution was stirred at room temperature for 20 h. Then, the reaction mixture was filtered to obtain the precipitated orange solids (1,1′-dilithioferrocene·(tmeda)$_2$), which was further washed with hexane. 9,9-dichloro-9-silafluorene (**2**, 699 mg, 2.66 mmol) was added to a THF solution (7 mL) of the obtained solids at −78 °C. After stirring the reaction mixture for 15 h and warming it to room temperature, all volatiles were removed under reduced pressure and hexane was added. After all insoluble inorganic salts were removed by filtration through a pad of celite, the solvent was removed from the filtrate under reduced pressure to obtain a red solid that was washed with $Et_2O$ to obtain sila[1]ferrocenophane **1** (648 mg, 1.78 mmol, 67%) as red crystals; **1**: red crystals, Mp = 190 °C (decomp.). $^1H$ NMR (400 MHz, $C_6D_6$) δ 7.76 (dd, $^3J$ = 7.4 Hz, $^4J$ = 1.3 Hz, 2H), 7.68 (dd, $^3J$ = 7.4 Hz, $^4J$ = 1.3 Hz, 2H), 7.30 (ddd, $^3J$ = 7.4 Hz, $^4J$ = 1.3 Hz, 2H), 7.22 (ddd, $^3J$ = 7.4 Hz, $^4J$ = 1.3 Hz, 2H), 4.49 (dd, $^3J$ = 3.6 Hz, $^4J$ = 1.8 Hz, 4H), 4.35 (dd, $^3J$ = 3.6 Hz, $^4J$ = 1.8 Hz, 4H); $^{13}C\{^1H\}$ NMR (101 MHz, $C_6D_6$) δ 148.2 (C), 134.5 (C), 134.0 (CH), 131.5 (CH), 128.2 (CH), 121.8 (CH) 78.6 (CH), 75.5 (CH), 28.1 (C); $^{29}Si\{^1H\}$ NMR (79.5 MHz, $C_6D_6$) δ −15.1. HRMS (DART), *m/z*: Found: 364.03478 ([M]$^+$), Calcd. for $C_{22}H_{16}SiFe$ ([M]$^+$): 364.03710. UV/vis (THF): $\lambda_{max}$ = 277 nm ($\varepsilon$ = 1.2 × 10$^4$), 476 nm ($\varepsilon$ = 1.9 × 10$^2$).

### 3.4. Thermal Polymerization of Sila[1]ferrocenophane **1**

A small amount of sila[1]ferrocenophane **1** was placed in an NMR tube attached with J-Young cock, and the NMR tube was degassed and heated at 190 °C for 1.5 h. After cooling at room temperature, orange solids of polymer **3**, which are almost insoluble in common organic solvents, such as THF, benzene, $CHCl_3$, and $CH_2Cl_2$, were obtained by filtration. The $^1H$ NMR spectrum of the suspension of the obtained solids is shown in Figure 5.

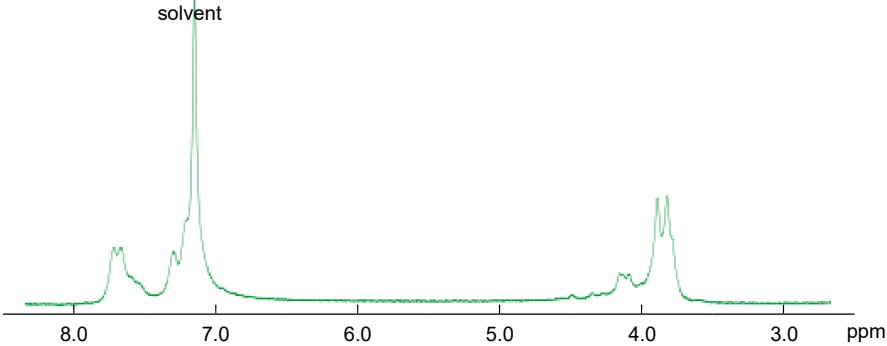

**Figure 5.** $^1H$ NMR spectrum of **3** in $C_6D_6$.

### 3.5. Reduction of Sila[1]ferrocenophane 1

In a glove box filled with argon, sila[1]ferrocenophane **1** (54.7 mg, 0.150 mmol) and $KC_8$ (11.7 mg, 0.299 mmol) were placed in a glass bottle. After adding 2 mL of THF to the glass bottle, the mixture was stirred for 10 h at room temperature. The resulting black suspension was filtered through a pad of celite. Bromoethane (1.0 mL, 12.8 mmol) was added to the filtrate at room temperature, and the reaction mixture was stirred for 5 h at room temperature. The volatiles were removed from the reaction mixture, and toluene was added to the mixture. After the filtration of the toluene suspension through a pad of celite, the solvent was removed from the filtrate under reduced pressure to obtain orange solids of **5**, which are almost insoluble in common organic solvents such as THF, benzene, $CHCl_3$, and $CH_2Cl_2$. The obtained solids were subjected to column chromatography [$SiO_2$, hexane] to obtain a small amount of **6**; **6**: orange solids, Mp = 100–101 °C. $^1$H NMR (400 MHz, $CDCl_3$) δ 7.86 (dd, $^3J$ = 7.4 Hz, $^4J$ = 1.2 Hz, 4H), 7.76 (dd, $^3J$ = 7.4 Hz, $^4J$ = 1.2 Hz, 4H), 7.43 (ddd, $^3J$ = 7.4 Hz, $^4J$ = 1.2 Hz, 4H), 7.33 (ddd, $^3J$ = 7.4 Hz, $^4J$ = 1.2 Hz, 4H), 4.68 (dd, $^3J$ = 3.4 Hz, $^4J$ = 1.8 Hz, 4H), 4.51 (dd, $^3J$ = 3.4 Hz, $^4J$ = 1.8 Hz, 4H); $^{13}$C{$^1$H} NMR (101 MHz, $CDCl_3$) δ 147.5 (C), 134.5 (C), 132.8 (CH), 131.3 (CH), 128.0 (CH), 121.1 (CH) 74.6 (CH), 72.6 (CH), 65.8 (C); $^{29}$Si{$^1$H} NMR (79.5 MHz, $CDCl_3$) δ −11.0. HRMS (DART), $m/z$: Found: 560.07190 ([M]$^+$), Calcd. for $C_{34}H_{24}Si_2FeO$ ([M]$^+$): 560.07158.

### 3.6. X-ray Crystallographic Analysis of 1 and 6

Single crystals of **1** and **6** were obtained after recrystallization from hexane. Intensity data for **1** and **6** were collected on the BL02B1 beamline of SPring-8 (proposal numbers: 2022A1200, 2022A1354, 2022A1584, 2022A1626, 2022A1705, 2022B1626, 2022B0589, 2023A1539, 2023A1771, 2023A1785, 2023A1794, 2023A1859, 2023A1925, 2023B1675, 2023B1806, and 2023B1878) on a PILATUS3 X CdTe 1M camera using synchrotron radiation (λ = 0.4135 Å). The structures were solved using SHELXT-2018 and refined using a full-matrix least-squares method on $F^2$ for all reflections using SHELXL-2018 [18]. All non-hydrogen atoms were refined anisotropically, and the positions of all hydrogen atoms were calculated geometrically and refined as riding models. Supplementary crystallographic data were deposited at the Cambridge Crystallographic Data Centre (CCDC) under deposition numbers CCDC-2305774 (**1**) and CCDC-2305775 (**6**); these can be obtained free of charge via www.ccdc.cam.ac.uk/data_request.cif.

### 3.7. Electrochemical Measurements

Cyclic and differential-pulse voltammograms were recorded on an ALS 1140A potentiostat/galvanostat (BAS Co., Ltd., Tokyo, Japan) using Pt wire electrodes under an argon atmosphere in custom-tailored glassware. Voltammograms were recorded at room temperature on $CH_2Cl_2$ solutions ([analyte]: 2.0 mM; supporting electrolyte: 0.1 M [$n$Bu$_4$][PF$_6$]) using a variety of scan rates.

### 3.8. Measurements of UV-Vis Spectra

UV-Vis spectra of 9-silafluorenylidene bridged sila[1]ferrocenophane **1** were recorded on a SHIMADZU UV-3150 UV-Vis-NIR spectrometer (Shimadzu Corp., Kyoto, Japan) under an argon atmosphere in 1 cm quartz cells.

### 3.9. Theoretical Calculations

Theoretical calculations for the geometry optimization and frequency calculations of **1** and **1′** were carried out using the *Gaussian 16* (Revision C.01) program package [19]. Geometry optimizations were performed at the B3PW91-D3(BJ) [20,21] level of theory using the 6-311G(3d) basis sets. Minimum energies for the optimized structures were confirmed by frequency calculations. NBO calculations were performed using NBO 7.0 program [22]. Computational time was generously provided by the Supercomputer Laboratory at the Institute for Chemical Research (Kyoto University). The coordinates of the optimized structures are included in the corresponding .xyz files as supporting information.

## 4. Conclusions

In summary, sila[1]ferrocenophane **1** was successfully synthesized as air/moisture-stable orange crystals. The 9-silafluorenylidene and ferrocenyl moieties were found to be aligned perpendicularly. On the basis of structural parameters both experimentally and theoretically obtained, **1** should be highly strained relative to the case of Ph$_2$Si-bridged sila[1]ferrocenophane **1′**, where the strain energies of **1** and **1′** were estimated as ca. 33.1 and 27.8 kcal/mol. Sila[1]ferrocenophane **1** was found to undergo ROP at a lower temperature relative to that of the SiPh$_2$-bridged sila[1]ferrocenophane (**1′**) to give the insoluble orange solids, which was probably due to the higher strain energy. Furthermore, the reduction of **1** also afforded the corresponding anionic polymer by the treatment of KC$_8$. A detailed investigation on the identification and properties of the obtained polymers is currently in progress.

**Supplementary Materials:** The following supporting information can be downloaded at https://www.mdpi.com/article/10.3390/inorganics12030066/s1, $^1$H and $^{13}$C NMR spectra of sila[1]ferrocenophane **1** and theoretically optimized coordinates (xyz) are available in the Supplementary Materials. Figure S1: $^1$H NMR spectrum of **1** in C$_6$D$_6$; Figure S2: $^{13}$C{$^1$H} NMR spectrum of **1** in C$_6$D$_6$; Figure S3: $^{29}$Si{$^1$H} NMR spectrum of **1** in C$_6$D$_6$.

**Author Contributions:** Conceptualization: T.S.; resources and funding acquisition: T.S.; experiments and data acquisition: S.U., K.S. and S.M.; writing—original draft preparation: T.S. and S.U.; writing—review and editing: T.S., S.U., K.S. and S.M.; project administration: T.S. All authors have read and agreed to the published version of the manuscript.

**Funding:** This work was financially supported by JSPS KAKENHI grants (23H01943, 22K18332, and 21KK0094) from MEXT (Japan), the Collaborative Research Program of the Institute for Chemical Research at Kyoto University (2023-13), a project subsidized by the New Energy and Industrial Technology Development Organization (NEDO), as well as JST CREST grant JPMJCR19R4.

**Data Availability Statement:** The raw data supporting the conclusions of this article will be made available by the authors on request.

**Acknowledgments:** We acknowledge the Supercomputer Laboratory in the Institute for Chemical Research of Kyoto University for the resources used. We would like to thank Toshiaki Noda for the expert manufacturing of custom-tailored glassware.

**Conflicts of Interest:** The funders had no role in the design of the study; in the collection, analysis, or interpretation of data; in the writing of the manuscript; or in the decision to publish the results.

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

Polymerizable Cyclic Ferrocenylsilane Fe(h-C$_5$H$_4$)$_2$(SiMe$_2$) with That of the Cyclic Ferrocenyldisilane Fe(h-C$_5$H$_4$)$_2$(SiMe$_2$)$_2$. *Organometallics* **1993**, *12*, 823–829.

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
