# Peer review of "[1]Ferrocenophane Bridged by a 9-Silafluorenylidene Moiety"

_inorganics, doi:10.3390/inorganics12030066_

Round 1
Reviewer 1 Report
Comments and Suggestions for Authors
The manuscript by Sasamori et al. reports a remarkable sila[1]ferrocenophane in which the silylidene unit is likewise embedded into a silafluorene moiety. In this title compound the HOMO is located at the bent ferrocenylene unit whereas the LUMO is mainly located at the silafluorene but also involves the iron atom according to the presented calculations. This feature is fully consistent with the electrochemical properties of the ferrocenylene unit and the UV-vis absorption. The ring tilt derived from the XRD structure is in the range of previously published other sila[1]ferrocenophanes and similarly the title compound can be ring opened thermally. I am convinced that these results alone are very attractive, scientifically sound, and worthy of publication.
I am not fully convinced whether the presence of compound 6 is actually necessary and helpful for the overall discussion. Its suggested formation seems somewhat vague. The hypothesis that air and moisture might be responsible is not fully plausible, cf. e.g. OM, 2000, 19, 2724 where a silafluorenyl anion is shown to be protonated which should be the fate of 4 with water as well. Moreover, compound 6 contains two silyl units per ferrocene whereas the anticipated precursors, be it monomeric or oligo-/polymeric, contain only one silyl unit per ferrocene and no evidence for desilylated derivatives is presented.
Minor typos:
Page 3, l.99: around the aryl groups
Page 7, ll. 238-240: sentence needs rephrasing.
Page 8: globe box-> glove box
Overall, an excellent paper which should be accepted after the minor changes outlined.
Author Response
Dear reviewer 1,
thank you very much for very kind comments. The reply is attached as a pdf file.
With warmest regards,
Takahiro

Reviewer 2 Report
Comments and Suggestions for Authors
Dear editor,
The manuscript reported the synthesis, structures and DFT calculations of Silaferrocenophane and 9,9-dichloro-9-silafluorene. I think such organometallic compounds are relatively interesting and have potential applications. I recommend the publication of the manuscript after some revisions. First, the two crystal structures in the manuscript are first reported and not reported elsewhere or not? This should be clearly clarified in the manuscript. In addition, the crystal descriptions of the crystal structures is too simple, I recommend the authors compare the two compounds with those similar compounds reported in literature and tell the readers why the two compounds are interesting.
Author Response
Dear reviewer 2,
thank you very much for very kind comments. The reply is attached as a pdf file.
With warmest regards,
Takahiro

Reviewer 3 Report
Comments and Suggestions for Authors
This is an interesting paper that is well written and has been prepared to a high standard. The supporting information is adequate.
I recommend acceptance subject to implementation of the following mostly minor changes:
Line 16 – change “were” to “was”
Line 56 – delete the word “was”
Line 57 – change “as orange solids” to “as an orange solid”
Line 67 – change “see the” to “see from the”
Lines 148/149 – change “its identification has not been undoubtedly identified” to “its identity has not been unambiguously established”
Line 245 – change “globe box” to “glove box”
Lines 294/295 – It should also be mentioned that 1H and 13C NMR spectra of cpd 1 are in the S.M.
Comments on the Quality of English Language
Mostly good - just a few changes needed.
Author Response
Dear reviewer 3,
thank you very much for very kind comments. The reply is attached as a pdf file.
With warmest regards,
Takahiro
